# Quality Control of Red Blood Cell Solutions for Transfusion Transported via Drone Flight to a Remote Island

**Koki Yakushiji [1]**, **Fumiatsu Yakushiji [2,3,]\***, **Takanori Yokochi [4]**, **Mikio Murata [5]**, **Michiyo Nakahara [6]**, **Naoki Hiroi [7]** and **Hiroshi Fujita [8]**

1  Graduate School of Media and Governance, Keio University, 5322 Endo, Fujisawa-shi, Kanagawa 252-0882, Japan; kokiy@sfc.keio.ac.jp
2  Department of Internal Medicine, Tokyo Metropolitan Bokutoh Hospital, 4-23-15 Koutoubashi, Sumida-ku, Tokyo 130-8575, Japan
3  Toho University School of Medicine, 5-21-16 Oomorinishi, Oota-ku, Tokyo 143-8540, Japan
4  Japan Airlines Co., Ltd., 2-4-11 Higashi Shinagawa, Shinagawa-ku, Tokyo 140-8637, Japan; yokochi.tp6x@jal.com
5  Department of Clinical Pharmacy, Yokohama University of Pharmacy, 601 Matanocho, Totsuka-ku, Yokohama-shi, Kanagawa 246-8589, Japan; mikio.murata@gmail.com
6  Clinical Laboratory, Tokyo Metropolitan Bokutoh Hospital, 4-23-15 Koutoubashi, Sumida-ku, Tokyo 130-8575, Japan; bo_kensa_yuketsu@tmhp.jp
7  Center for Medical Education, Faculty of Medicine, Toho University, 5-21-16 Oomorinishi, Oota-ku, Tokyo 143-8540, Japan; n-hiroi@med.toho-u.ac.jp
8  Department of Transfusion Medicine, Tokyo Metropolitan Bokutoh Hospital, 4-23-15 Koutoubashi, Sumida-ku, Tokyo 130-8575, Japan; hiroshi_fujita@tmhp.jp
*  Correspondence: clinic@nifty.com; Tel.: +81-3-3633-6151

**Abstract:** Long-distance transoceanic transport of blood using drones has never been reported. This study aimed to prove that blood transportation via drones can meet the rapid demand for blood transfusions anywhere in Japan, including remote islands. We demonstrated the transport of red blood cells (RBCs) packs using a drone over the sea from Sasebo to Arikawa port. Drone operations were conducted visually only at take-off and landing. Cruise flights were conducted via satellite-based remote control from Tokyo. The RBC solutions were transported at 2–6 °C to avoid hemolysis. Hemolysis was assessed visually and by measuring lactate dehydrogenase (LDH) levels before departure and upon arrival at Tokyo Metropolitan Bokutoh Hospital to evaluate whether RBCs were transfusable. LDH levels of the RBC solutions before and after transport were 57.5 ± 3.1 vs. 64.0 ± 2.9. RBC solutions were transported via air and land from Tokyo to Sasebo and showed no remarkable signs of hemolysis. Remote RBC solution transport by uncrewed helicopters with temperature control is feasible and allows RBC transportation in emergencies involving disrupted land transportation, such as the COVID-19 pandemic.

**Keywords:** blood transfusion; drone; seas; transportation; COVID-19; temperature; unmanned helicopter

## 1. Introduction

RBC solution transportation by small, unmanned aerial vehicles (drones) was first performed in the United States by Amukele et al. [1] and has been an ongoing practice in Rwanda [2–4]. Long-distance transport experiments have been conducted in the United States [5] with fixed-wing drones, where a pack of red blood cell (RBC) solution is dropped from the drone at the destination site. A previous report of our authorship revealed that dropping of RBC solutions causes blood deterioration and carries significant problems [6]. There have been no conclusive reports about the condition of blood in Rwanda following drone transport. Instead of throwing blood packs down from fixed-wing drones in the air, multi-copters allow a safer delivery. Therefore, multi-copter transport would be an

appropriate choice to prevent blood degradation. We and Okada et al. have previously conducted basic experiments on blood transport using multi-copters but were limited to visual flights and did not involve unseen flights [7,8]. Moreover, this is not feasible for transoceanic transport without terrestrial cable installation. Therefore, transoceanic flights are only feasible using non-visual flights via satellite communications. Consequently, there have been no reports of successful RBC solution transportation using multi-copters or autonomous helicopters.

On the other hand, there have been many reports on emergency blood transport by manned conventional helicopters and a review of reports to consider actual helicopter emergency transport [9,10]. In addition, helicopter blood transfusion transport is now allowed in New York City, where it had not been allowed before [11].

Recent reports indicate that vertical take-off and landing aircraft type drones have been used to transport blood products over long distances and to isolated islands in Lake Victoria [12].

In areas in Japan with a large population, a stable supply of blood for transfusion is needed to ensure that the blood stored in hospitals is maintained and the additional requirement for blood is met, especially when the demand for blood is high. This system provides an additional supply chain for the quick transport of blood to the hospital. Blood transportation in Japan is via automobiles and in the absence of a land traffic blockage caused by a particular disaster, the supply of blood is stable by car. We have also found no blood deterioration during transport by automobile [13].

Japan has many remote islands inhabited by people in the western Pacific Ocean. Most of these islands have medical facilities and encounter events requiring blood transfusions. There has been an effective supply of blood demonstrated by the day-long transport of RBC solution by ship as far as 1000 km away in the Ogasawara Islands [14,15], where a stable system has been established for normal RBC solution supply while maintaining the quality of the RBC solution. However, the transportation of an RBC solution by aircraft during an emergency has not been adapted.

A distance of 50 km is considerable, especially during an emergency, and blood transport by ship can be affected by high waves. The trial we conducted is a novel approach for supplying blood that cannot be accomplished by sea transport. We investigated the feasibility of using a remotely operated, uncrewed helicopter (UH) for the transport of RBC solution over a distance of 55,000 m to a remote island of Japan.

Therefore, this study aimed to prove that blood transportation via drone can meet the rapid demand for blood transfusions anywhere in Japan, including remote islands. In addition, we conducted a demonstration experiment to ensure that the quality of unused transported RBC solution can be maintained and used up to the 3-week expiration date in Japan with correct management.

## 2. Materials and Methods

### 2.1. Equipment for Flying and Materials for RBC Solution Quality Preservation

The UH used was FAZER RG2 (Yamaha Motor Co., Shizuoka, Japan). The following specifications based on Japanese type certification were as follows: dimensions, $3664 \times 734 \times 1226$ mm (including the rotor); bodyweight of the aircraft, 79.5 kg; maximum take-off weight, 110 kg; cruising range, 90 km; loading weight, 35 kg; cruising time, 100 min; maximum speed, 72 km/h. The weight of the luggage on the two flights with mixed luggage was 18 kg. Temperature stabilizers in boxes were used for RBC transportation during the flight. One was a pair of RBC Constar IIs (polyethylene glycol-based refrigerant; freezing temperature, 4 °C; dimensions, $13 \times 24 \times 2.5$ cm; Daido Industries Inc., Osaka, Japan) in a shipping box called NeoAce (Sunrex Industry Co., Ltd. Yokkcichi, Mie, Japan; dimensions, $36.5 \times 28 \times 30$ cm; weight, 1.4 kg (including synthetic fiber chips to fill the gaps; hereinafter Box A), usually used by the Japan Red Cross (JRC), which was placed in an environment approximately equal to the outside temperature on a round-trip flight. Box A is made up of a synthetic fiber with an aluminum cover outer box, urethane

cushioning material, which is also with an aluminum cover, within which synthetic fiber in light polyethylene bags fills the gaps between the packs of RBCs.

The other was a pair of RBC cooler type IIIs (paraffin-based refrigerant; freezing temperature, 4.5 °C; dimensions, 15 × 23 × 2 cm; JSP Co., Ltd., Tokyo, Japan) in a transport heat shield box (Utsunomiya Industry Co., Aichi, Japan; dimensions, 25 × 25 × 21 cm; weight, 320 g (with an air-filled plastic and a certain foil covering on the outside; hereinafter Box B), which was used only on the outward journey and placed in hot conditions near the exhaust muffler of the UH (Figure 1).

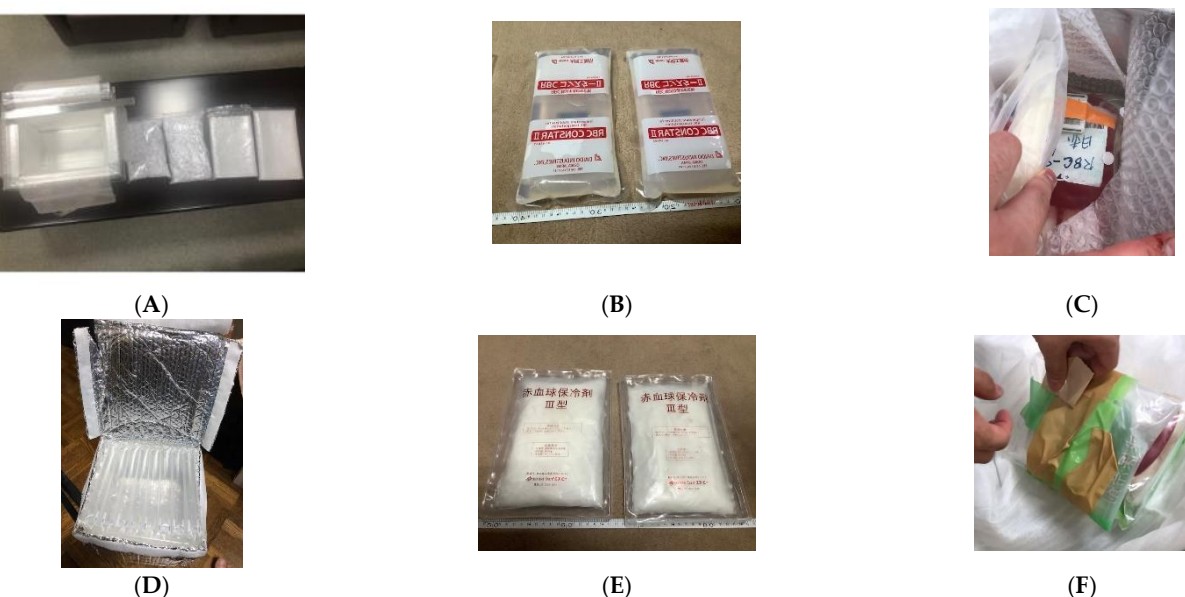

**Figure 1.** Blood transport equipment without active transport refrigerator. (**A**) Box A (NeoAce) and four refill chip bags for space; (**B**) Constar II; (**C**) RBC solution sandwiched by Constar II used for box A; (**D**) Box B. Transport heat shield box (Ustunomiya); (**E**) Type III; (**F**) RBC solution sandwiched by the type III used for box B.

ATR700 and ATR705 (active transport refrigerator, Fujifilm Toyama Chemical Co., Ltd., Tokyo, Japan) are devices that can transport RBC solutions at 2–6 °C regardless of external temperature. ATR700 is an RBC solution transfer device, while ATR705 is used for static RBC solution storage and temperature control.

We recorded the surface temperatures of the RBC solution pack using an electronic watch logger, which records the temperature every minute (KT-265F; Fujita Electric Works, Ltd., Ninomiya, Kanagawa, Japan), and the vibration of UH movement was recorded using a 3-Axis Vibration datalogger, DT-178A (frequency range, 0–60 Hz; Shenzhen Everbest Machinery Industry Co., Ltd., Shenzhen, China).

### 2.2. UH Operation Methods and Operational Responsibilities

This flight was conducted with the permission of the Osaka Regional Civil Aviation Bureau (ORCAB), the Ministry of Land, Infrastructure, Transport, and Tourism (MLITT). When Japan Airlines (JL) applied for permission from the ORCAB based on the "Approval Procedures for Unmanned Aircraft Flight Permits" stipulated by the MLIT [16], we notified in advance the date, time, and flight path (same as the result route in brief) to the manned aircraft operating organizations (All Nippon Airways Federation and six other organizations) as the flight was to be conducted "out-of-sight without an assistant" and JL applied for the permission for "transport of dangerous goods" due to the transport of sputum samples for antigen test specimens of a virus in another UH operation.

The ORCAB imposed weather restrictions of <10 m/s wind speed and 5 mm rainfall.

The flight was performed by the pilots at JL. They were in constant communication with the local pilots at Sasebo and Arikawa ports to assess the weather and wind speed. The UH was controlled by a local pilot under visual observation during take-off and landing

commonly used in Japan. As the ORCAB does not have permission to fly over third parties, flying over navigating ships by hovering to wait risks must be avoided.

The cruise flight at an altitude of 100 m was performed by pilots at JL. The UH was controlled via satellite telephone communication (Inmarsat). A telephone line using a communications satellite can transmit control communication to a UH in flight, even at sea; thus, the UH could be operated by a pilot at the JL office in Tokyo. The control of the UH flight was based on Inmarsat technologies and Yamaha's technologies [17].

Although the helicopter was not equipped with a specific collision avoidance system, the flight environment on that day was at an altitude of ≤150 m, which is restricted by drone regulations to avoid the risk of collision with aircraft and there was no overlapped airspace. There have been no other government-permitted drone operations flying <150 m.

### 2.3. Source and Production of RBC Solution Products

Five packs of irradiated RBC solutions (280 mL) commonly used in Japan for transfusion to prevent transfusion-associated graft versus host diseases (TA-GCVHD), which have been known to be lethal complications, were obtained from the JRC (Tokyo, Japan) (RBC solution type A: two donors, type B: one donor, type AB: two donors) and were labeled #1 to #5. The RBC solution (280 mL) was derived from leukocyte-depleted whole blood (400 mL) packed in a citrate-phosphate-dextrose-adenosine-containing bag and was stored in a mannitol, adenosine, phosphate solution. It should be added that the hazard classification guide (last updated on 22 October 2012) clearly states that blood for transfusion is not a UN3373 blood sample and is not considered hazardous material [18].

The RBC solution was divided into two bags: a 250 mL RBC solution in the parent bag and a 30 mL RBC solution in a small volume bag (BB-TQ008J; Terumo Co. Ltd., Tokyo, Japan). The 30 mL RBC solution was stored in an RBC solution bank refrigerator (MBR-107T(H); SANYO Electric Co., Ltd., Osaka, Japan) at Tokyo Metropolitan Bokutoh Hospital (TMBH) and served as the control group. To track the movement of RBC solution packs and their origin, branch number 1 was assigned to RBC solutions which made the trip and branch number 2 was assigned to the controls. Particularly, #5-2 was separated equally into two RBC solution packs and numbered with the original #5-2 and #5-3; #5-3 was the control and #5-2 made the trip.

The packaging of blood was similar to that use for drugs and not for hazardous materials. The packs were always transported in clear, sealable plastic bags to prevent blood spillage, as it was not necessary to open them for demonstration trial, in case the packaging was damaged during transport. In addition, all transport boxes were hollow and designed to prevent blood spillage.

### 2.4. RBC Solution Transportation Schedule

On day-1, the packs of RBC solution inside a plastic bag in an ATR700 were transported by automobile from TMBH to Haneda airport in Tokyo, 30 km away. The ATR700 was then transferred to an aircraft (Boeing 737) and transported over 950 km to Nagasaki airport, followed by an automobile transfer of 80 km to Sasebo City. In the automobile, the ATR700 was placed on the back seat, whereas in the airplane, it was placed next to the economy class seat. On day 1, the packs of RBC solutions were transported over 55 km by the UH (FAZER RG2) to Arikawa port (Figure 2). The packs of RBC solution in the ATR700 or in box A or B were placed on a boat-like platform suspended from the UH. The vibration and surface temperature of the packs of RBC solution were recorded on the outward journey. The packs were then transported by automobile on the back seat for approximately 2 km to Kamigoto Hospital. On the return on day 2, the packs were transported back to Sasebo city and on day 3 back to TMBH in the same way as during the outward journey.

**Figure 2.** Flow diagram of the round-trip trial for RBC solution transportation for transfusion.

The upper side panel shows the method of RBC solution transport. The lower panel shows the procedures of the trial. On day 1 at Tokyo Metropolitan Bokutoh Hospital, we divided packs of RBC solution for storage and the round-trip trial and sampled for controls. On day 1 at Sasebo, we prepared for UH transportation and stored the RBC solution for a second control. At Kamigoto Hospital on day 2, we prepared for the round trip of the uncrewed flight, and the RBC solution quality was evaluated.

*2.5. RBC Solution Operations for Transport*

On day 1 at TMBH, the RBC solution samples were divided for use in storage, round-trip trial preparation, and samples for controls (Figure 2, lower panel). Stored packs of RBC solution were labeled #1-2, #2-2, #3-2, #4-2, and #5-3, while transported RBC solutions were labeled #1-1, #2-1, #3-1, #4-1, #5-1, and #5-2. The RBC solutions were transported to Sasebo and stored overnight.

On day 1 at Sasebo, #1-1 and #2-1 were stored in the ATR700, #3-1 was sandwiched by Constar IIs in box A, and #5-2 was sandwiched by type IIIs in box B for the UH flight to Arikawa port. Upon arrival at Kamigoto Hospital, #1-1, #2-1, #3-1, and #5-2 were stored in the ATR700 overnight, and #4-1 and #5-1 were stored overnight in the ATR705 at Sasebo as second controls.

On day 2, the RBC solutions in the ATR700 were transported by car to Arikawa port. For the return UH flight, #1-1, #2-1, and # 5-2 were stored in the ATR 700, and #3-1 was sandwiched by Constar IIs in box A. Upon arrival at Sasebo, #3-1 was placed into ATR 700 after the flight was stored overnight in the ATR 700.

On day 3, RBC solutions #4-1 and #5-1 were stored in the ATR 705 at Sasebo, placed into ATR 700, and all the RBC solutions in the ATR 700 were returned to Tokyo by the same transportation as the outbound trip. After the round trip, we evaluated the quality of the RBC solutions.

*2.6. Temperature and Vibration Measurements*

2.6.1. Temperature

To measure temperature, we used the ATR700's automatic measurement of the inside and outside temperatures. It started recording before the take-off of the UH and stopped after the ATR700 returned to TMBH.

The surface temperatures of #3-1 and #5-2 were measured with electronic watch loggers fixed directly to the surface of each blood pack with adhesive tape.

In addition, the inside temperature of box B was measured by an electronic watch logger.

2.6.2. Vibration

To measure vibration, a vibration logger was attached to the inside of the cloth cover of the ATR700, which traveled the entire journey.

The vibration was measured from before the take-off of the UH on day 1 through the return to TMBH on day 3.

### 2.7. Evaluation of the Effects of Transport on RBC Solution Quality

We performed a hematological and macroscopic evaluation of the transported RBC solutions and controls by comparing the quality of the transported RBC solution in the control and study groups based on levels of lactate dehydrogenase (LDH), aspartate aminotransferase (AST), potassium, blood sugar, lactate, ammonia, and hematocrit. These were evaluated by Bio Medical Laboratories (BML Inc., Tokyo, Japan).

RBC solutions in the control and study groups were filtered using a transfusion set (Terumo Co. Ltd., Tokyo, Japan) (Figure 3), and the LDH and AST levels were measured in the RBC solutions post-filtration. LDH and AST were measured by Bio Medical Laboratories.

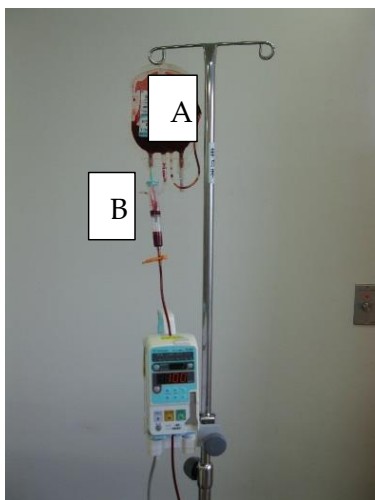

**Figure 3.** Red blood cell (RBC) solution and a transfusion set.

Four aliquots from each RBC solution sample were used for biochemical tests, and one sample was used for hematological tests. We evaluated biochemical hemolysis based on LDH levels. For each RBC solution sample, a test was performed for LDH with laboratory residues. Since AST and LDH are related, only an LDH test was performed. Statistical significance was set at 5% for comparison of means by *t*-test. All statistical calculations were performed using Excel version 2009. Data are expressed as group means $\pm$ standard deviation of the mean.

For cross-matching tests, the whole RBC solution from the transported segment tubes was cross-matched using a saline method and an indirect antiglobulin test.

From Figure 3, A represents a pack of RBC solution, and B indicates a transfusion set (Terumo Co. Ltd., Tokyo, Japan).

### 3. Results
### 3.1. Flight of UH
3.1.1. Time and Status

The UH flight carrying the RBC solution samples took place on 6 November 2020 from the coast of Sasebo City to Arikawa Port in Shinkamigoto Town. The return flight was on the following day (Table 1).

**Table 1.** Flight information.

| Date | Time and Calculated Ground Speed * | Weather, Temperature, Wind | Operation | Flight Section, Distance Weight of the Luggage | Uncrewed Helicopter, Cruise Speed Altitude |
|---|---|---|---|---|---|
| 6 November 2020 | 9:57–11:01; 63 min and 30 s 52 km/h | Cloudy; 16.6 °C; East wind 1.0 m/s at 10:00 in Sasebo. Cloudy; 21.9 °C; South-southeast wind 2.9 m/s at 11:00 in Arikawa. | Visual observation at take-off and landing point. Cruise, re-motely controlled by satellite link at JL headquarters: Out-of-sight without assistant. | Coast of Sasebo in Sasebo city to Arikawa port in Kamigoto city. Approximately 55 km. 18 kg. | FAZER RG2. Cruise: round 6000 rpm; air speed, 18 m/s; altitude 100 m |
| 7 November 2020 | 9:41–10:50; 69 min and 1 s 48 km/h | Light rain; 19.6 °C; North-northwest wind 2.9 m/s at 10:00 at Arikawa. Cloudy; 24.0 °C; East-northeast wind 1.5 m/s at Sasebo. | Visual observation at take-off and landing point. Cruise, remotely controlled by satellite link at JL headquarters: Out-of-sight without assistant. | Arikawa port to coast of Sasebo. Approximately 55 km 18 kg. | FAZER RG2. Cruise: round 6000 rpm; Speed, 18 m/s; altitude 100 m. |

\* Including takeoff and landing time, Abbreviations: JL, Japan Airlines co. Ltd.

Figure 4 shows the aircraft before take-off from Sasebo on its outward journey. The cruise flight took place as planned at an altitude of 100 m.

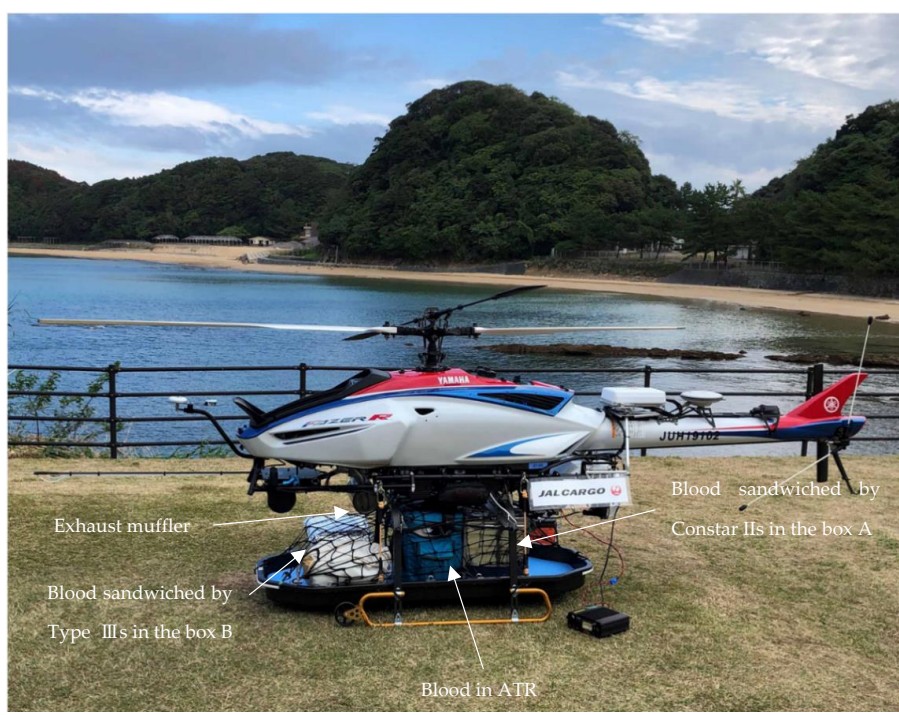

**Figure 4.** Industrial uncrewed helicopter (FAZER RG2) at the coast of Sasebo City just before the flight to Arikawa port.

RBC #3-1, which was sandwiched by Constar IIs in box A, was placed in the back row, #1-1 and 1-2 in the ATR700 were placed in the middle, and #5-2, which was sandwiched by type IIIs in box B, was placed on the right side of the front row close to the exhaust muffler. They were on a boat-like platform as shown. All the cargo on the platform was secured with rubber nets to prevent movement. The photo was taken at 9:45 on November 6.

We monitored the outbound flight in Sasebo. The actual take-off and landing were performed by local pilots under visual observation. During the cruise, the flight was

simultaneously monitored at JL and at Sasebo and Arikawa ports. Figure 5 shows UH flight management, including the airspeed of UH of approximately 15 m/s, engine revolution per minute of 6000, and flight direction of approximately 260 degrees.

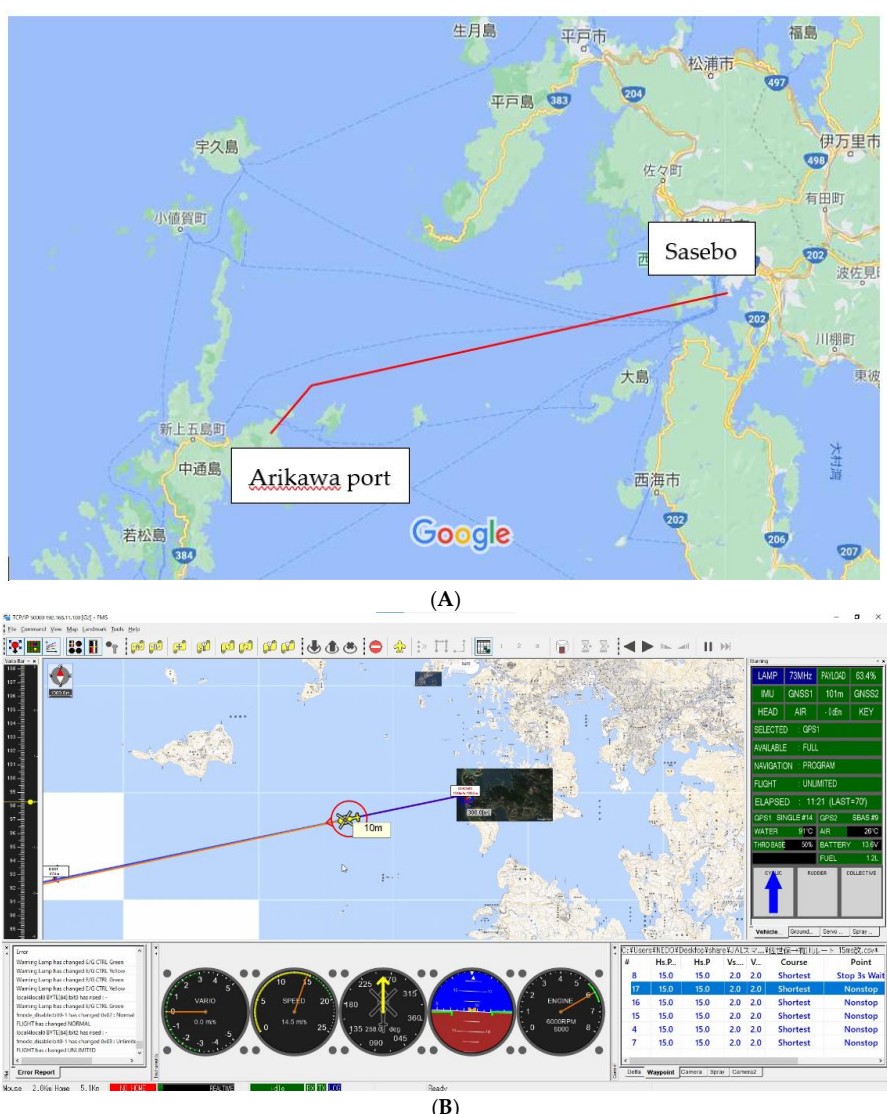

(**A**)

(**B**)

**Figure 5.** Uncrewed helicopter (UH) flight management. (**A**) Round-trip route between Sasebo and Arikawa Port. This shows the outline of the flight path in the flight application and how the actual flight was conducted. The flight distance was approximately 55 km one way; (**B**) The flight from Sasebo to Arikawa port on 6 November 2020. The airspeed of UH was approximately 15 m/s (54 km/h), the engine revolution was 6000 per minute, and the flight direction was approximately 260 degrees.

The flight time was different between the outward and return trips, mostly due to the wind speed and direction.

The containers for the RBC solution on the platform that UH was hanging from remained intact.

The flight time was different on the outward and return trips. Due to government approval, the airspeed on the return trip was limited to 72 km/h or less, so there was no way to recover from the wind-induced slowdown on the return trip. Figure 5 shows a copy of the actual screen at the Japan Airlines headquarters and an image of the monitor at take-off and landing points.

### 3.1.2. Temperature

As a function of the ATR700, a continuous temperature record from before the departure from TMBH to the return to TMBH is shown in Figure 6.

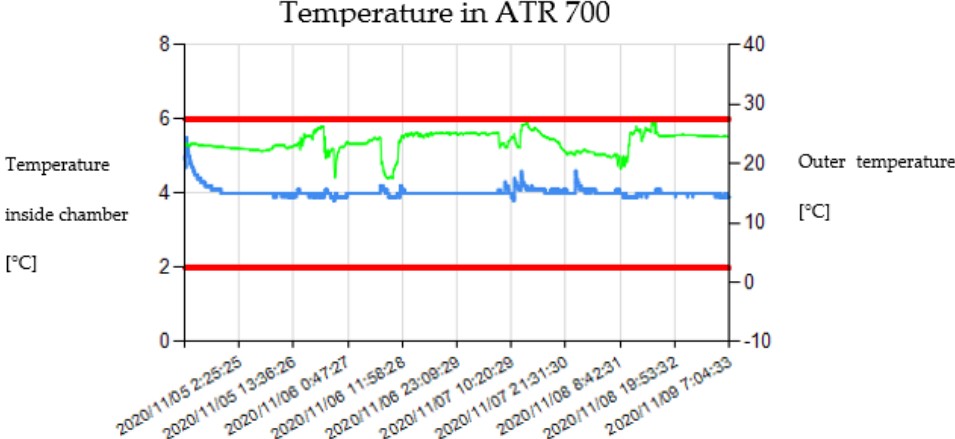

**Figure 6.** Temperature in ATR 700 during the trial.

The surface temperature of the blood pack during the UH flight is shown in Figure 7.

Blood packs #1-1 and #2-1 were flown from TMBH to Kamigoto Hospital and back, including a UH retaliatory flight in the ATR. In both cases, the temperature of the blood packs was at the appropriate level of 2–6 °C, and the temperature recorded by the ATR was also at the appropriate level (3.8–4.7 °C; Figure 6).

Blood packs #1-1 and #2-1 remained in the ATR 700. The temperature inside the chamber (blue line) was kept at 2–6 °C. The outer temperature was indicated by green. The temperature between the upper and lower red lines indicates safe temperature circumstances. The point of change in ambient temperature is where other blood packs were moved in and out of the ATR700 into other containers.

Figure 7 shows the temperature of the surface of #3-1 and #5-2. A and B show the levels of the surface temperature of #3-1 were appropriate at 2–6 °C.

Figure 7C shows the temperature level of the surface of #5-2, which was <6 °C; Although the temperature was below the standard of 6 °C, it continued to rise, and temperature control was inadequate. On the way back, #5-2 was in ATR 500 without type III.

The surface temperature of #5-2 increased, a result that seems to be related to the temperature increase caused by not blocking the outside and exhaust air temperatures related to the transport of the RBC solution using UH.

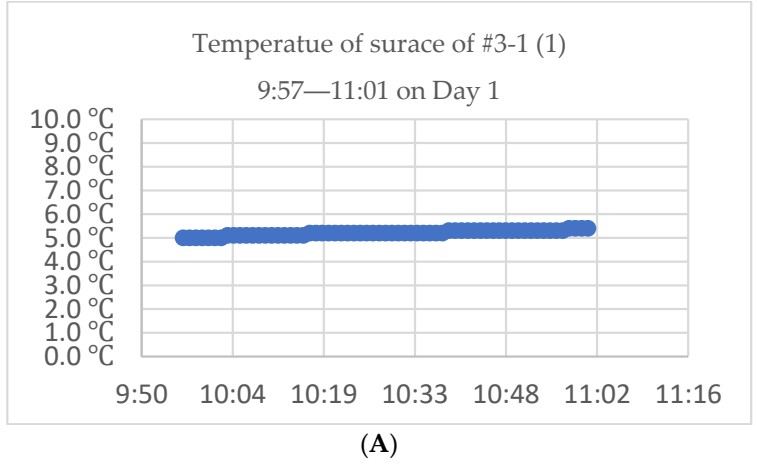

**Figure 7.** *Cont.*

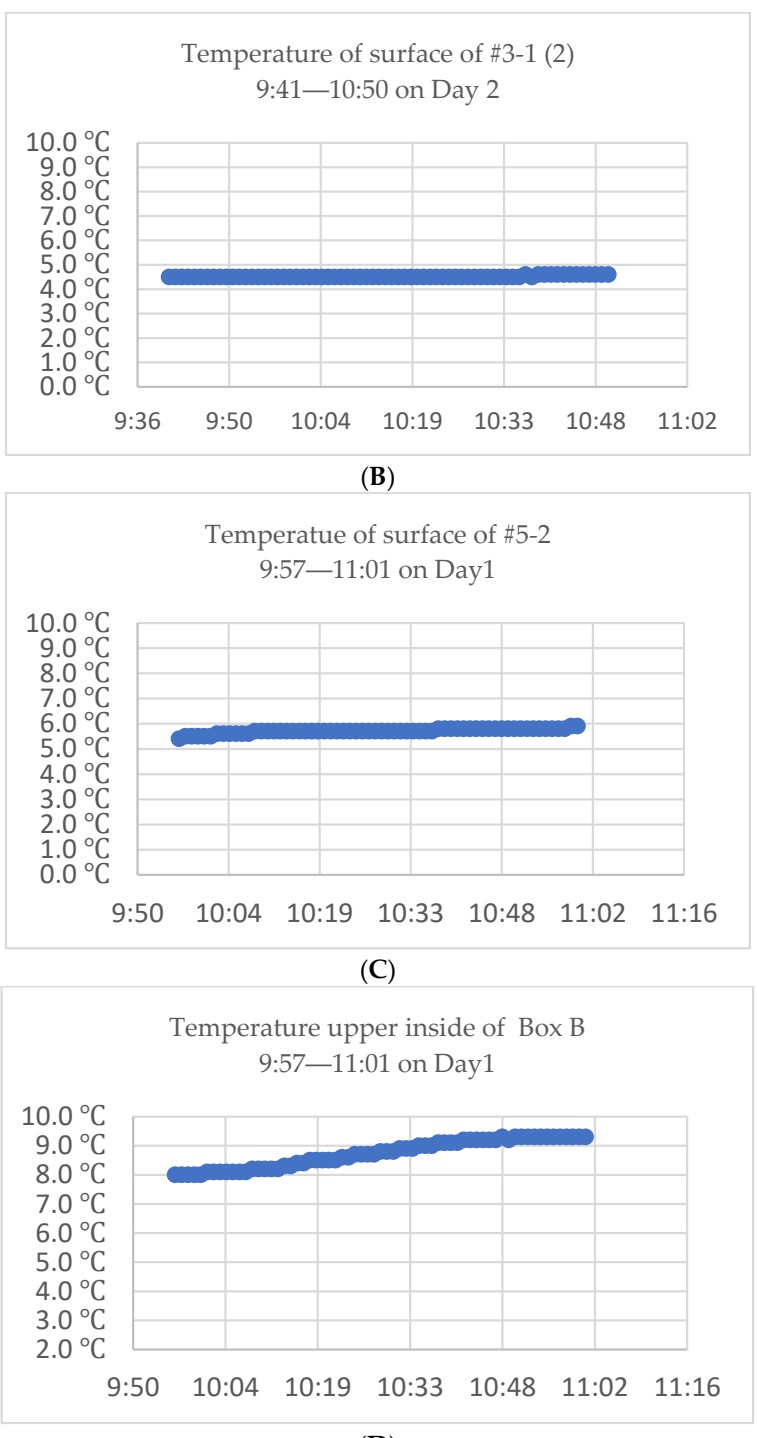

**Figure 7.** The surface temperature of RBC solution packs #3-1 and #5-2 on the flight from Sasebo to Arikawa port. (**A**,**B**) show the temperature of #3-1, which was sandwiched by ConstarIIs in box A on the round-trip flight. The flight times were 10:57 to 11:01 a.m. on day 1 (**A**) and 9:41 to 10:50 a.m. on day 2 (**B**); (**A**,**B**) shows the temperature levels of the outward flight and return at 2–6 °C; (**C**) shows the temperature of #5-2, which was sandwiched by type IIIs in box B during the outward. The surface temperature of #5-2 was below 6 °C; however, there was an increase, and it was not sufficient temperature control. In the return flight of #5-2, it was put in ATR 700 without type III; (**D**) shows the temperature of inside box B. The temperature was rising, probably due to outside and exhaust air temperatures.

### 3.1.3. Vibration

UH was measured on the journey to determine external forces (Figure 8). The external forces were calculated as follows:

$$External\ force(g) = \sqrt{X-axis\ force^2 + Y-axis\ force^2 + (|Z-axis\ force| - 1)^2}(g)$$

Figure 8 shows that the total external forces on the UH were slightly greater than those in an automobile. In addition, the vibration continued during the UH flight. This was because loggers could only measure low frequencies <60 Hz; therefore, high-frequency vibrations were not recorded. The large values when walking and opening the cloth case for opening/closing the ATR or checking operation were art (facts of the location of the logger attachment. We acknowledge the lack of detailed time records for the different stages of flight, e.g., take-off, landing, and cruise flight.

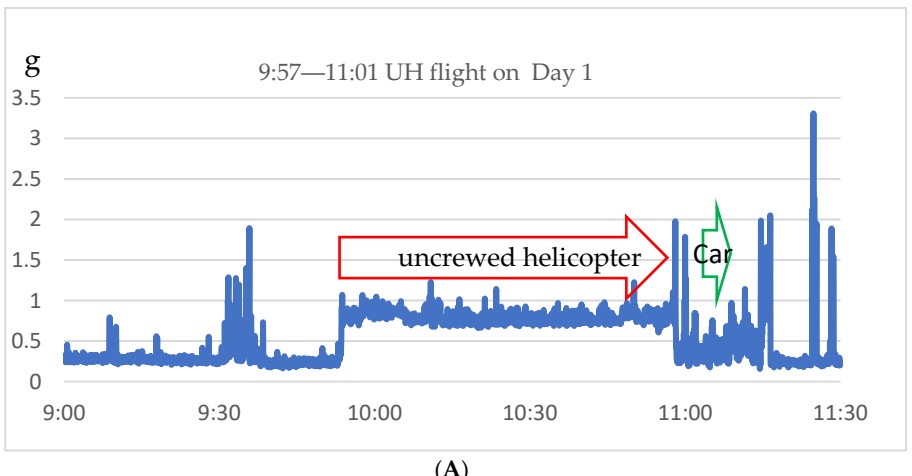

**(A)**

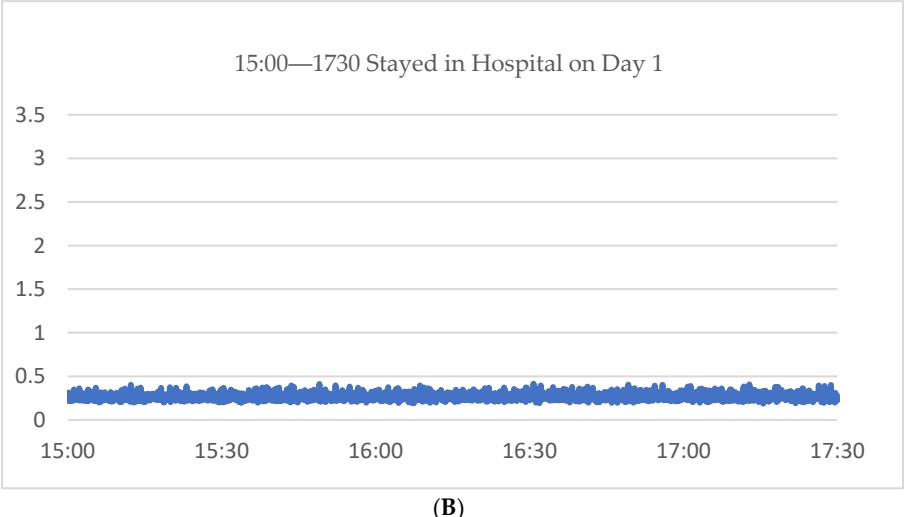

**(B)**

**Figure 8.** *Cont*.

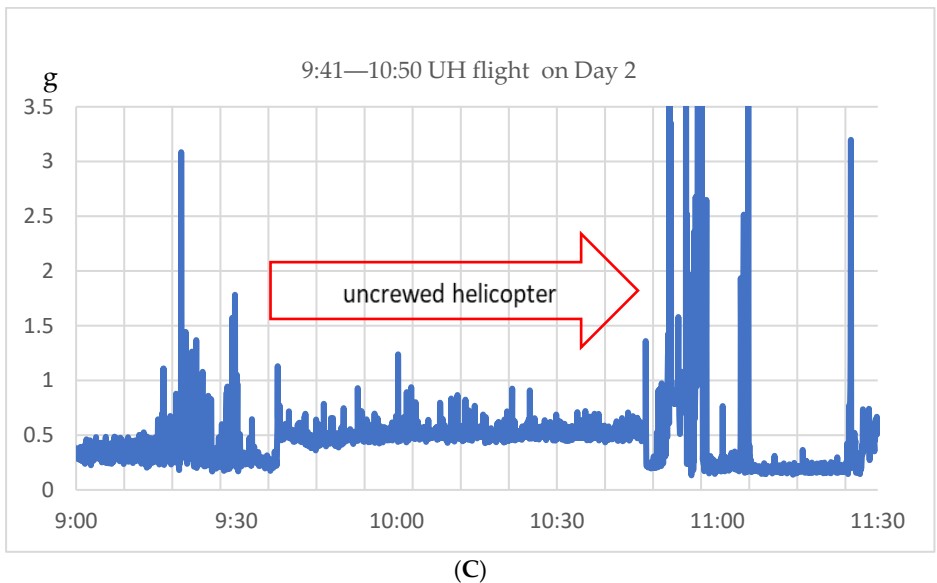

**(C)**

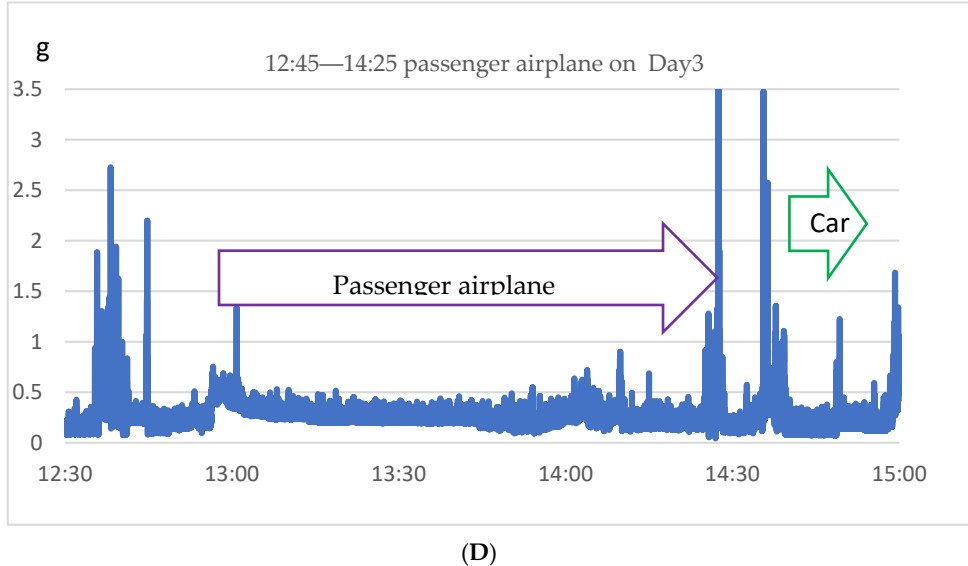

**(D)**

**Figure 8.** The graph shows the vibrations on the surface of the ATR cloth case. The external force was calculated as the force on the three-axis componentry as follows: $External\ force(g) = \sqrt{X-axis\ force^2 + Y-axis\ force^2 + (|Z-axis\ force| - 1)^2}(g)$ **(A)** shows the UH flight from Sasebo to Arikawa port in Kamigoto; **(B)** shows the ATR vibration staying at Kamigoto Hospital; and **(C)** shows the UH flight back to Sasebo; **(D)** is the return from Nagasaki Airport to Haneda Airport in Tokyo and the taxi to TMBH. The temporary total external force on the UH (red arrow) was slight greater than the external force on a car (green arrow). However, the vibration continued during the UH flight because the loggers could only measure low frequencies (<60 Hz); therefore, high-frequency vibrations were not recorded. High-frequency vibrations are considered to have a significant impact, and therefore, the effects of external forces could not be accurately determined. The large values when walking and opening the cloth case for opening/closing the ATR or checking operation were artifacts of the location of the logger attachment.

### 3.2. Macroscopic Findings and Hematological Evaluation

Figure 9 shows the macroscopic findings of the RBC solution. Figure 9A,B are macroscopic images of RBC solutions #1 and #2, which were transported in the ATR700 from Tokyo to Kamigoto Hospital. The RBC solutions on the left (#1-1 and #2-1) were the trans-

ported RBC solutions, and the RBC solutions on the right (#1-2 and #2-2) were stored at TMBH. No significant macro-hemolysis was observed.

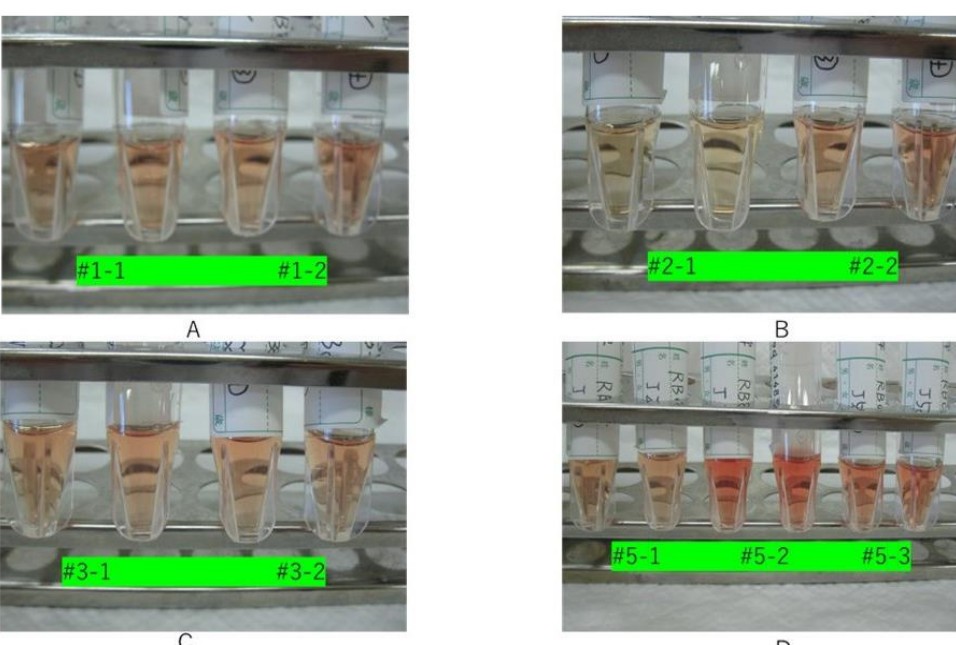

**Figure 9.** Macroscopic findings of RBC solution supernatant. (**A**,**B**) RBC solutions #1 and #2, which were transported by ATR700 from Tokyo to Kamigoto Hospital. #1-1 and #2-1 were transported RBC solutions, while #1-2 and #2-2 were stored at Tokyo Metropolitan Bokutoh Hospital (TMBH). No substantial macro-hemolysis was observed; (**C**) #3-1 was sandwiched by Constar IIs in box A from Sasebo to Kamigoto Hospital, and #3-2 was stored at Tokyo Metropolitan Bokutoh Hospital. There was no significant macro-hemolysis; (**D**) #5-2 and #5-3 appeared to have undergone more hemolysis than #5-1. Hemolysis was also suspected for #5-1. Although #5-3 was stored at TMBH, it was suspected to have undergone hemolysis.

Figure 9C shows that #3-1, which was sandwiched between Constar IIs in box A during the flight from Sasebo to Kamigoto Hospital, did not show significant macro-hemolysis.

Figure 9D shows the macroscopic findings of #5-1, which completed the round trip to Sasebo; #5-2, which was sandwiched between type IIIs in box B during the flight from Sasebo to Kamigoto Hospital; and #5-3, which was stored at TMBH. On visual observation, #5-2 and #5-3 were suspected of having undergone hemolysis compared with #5-1. Hemolysis was also suspected for #5-1. Although #5-3 was stored at TMBH, the results indicated possible hemolysis.

### 3.3. Hematological Data

Hematological data were collected before and after the round-trip transportation (Table 2).

### 3.4. Evaluation of Biochemistry

Biochemical tests were conducted before the trial and upon return to the laboratory after the experiment. The values of AST and LDH of RBC solution #2 were reversed before and after the experiment. The reason for this is unknown, but the large standard deviation may indicate that the pre-trial sampling was unreliable. In particular, for the LDH evaluation before the trial, we excluded the fourth sample from the analysis (Table 3). Blood glucose decreased over time, and AST and LDH showed corresponding changes.

**Table 2.** Hematological data before and after the round-trip transportation.

| Blood Type | # | RBC Solution Storage/Transport | WBC (/μL) | RBC (10$^6$/μL) | Hb (g/dL) | Hct (%) | MCV (fL) | MCH (pg) | MCHC (%) | Plt (10$^6$/μL) |
|---|---|---|---|---|---|---|---|---|---|---|
| A+ | 1 | Before trial | 340 | 6.72 | 19.7 | 56.8 | 85 | 29.3 | 34.7 | 0.4 |
| | 1-1 | Round trip to Kamigoto Hospital (ATR) | 300 | 6.80 | 19.9 | 56.6 | 83 | 29.3 | 35.2 | 0.2 |
| | 1-2 | No trip (stored) | 370 | 5.64 | 16.5 | 47.7 | 84 | 29.3 | 35.0 | 0.2 |
| A+ | 2 | Before trial | 230 | 7.14 | 23.4 | 66.5 | 93 | 32.8 | 35.2 | 0.3 |
| | 2-1 | Round-trip to Kamigoto Hospital (ATR) | 410 | 7.14 | 17.7 | 50.3 | 76 | 26.6 | 35.2 | 0.1 |
| | 2-2 | No trip (stored) | 240 | 7.30 | 23.6 | 67.4 | 92 | 32.4 | 35.0 | 0.9 |
| B+ | 3 | Before trial | 400 | 5.58 | 17.1 | 49.6 | 89 | 30.6 | 34.5 | 0.3 |
| | 3-1 | Round-trip to Kamigoto Hospital (ATR+Constar II) | 210 | 6.02 | 18.6 | 52.4 | 87 | 30.9 | 35.5 | 0.2 |
| | 3-2 | No trip (stored) | 290 | 5.75 | 17.4 | 50.9 | 89 | 30.3 | 34.2 | 0.5 |
| AB+ | 4 | Before trial | 270 | 6.52 | 20.7 | 5.79 | 89 | 31.8 | 35.8 | 0.3 |
| | 4-1 | Round-trip to Sasebo City (ATR) | 220 | 6.56 | 20.6 | 58.0 | 88 | 31.4 | 35.5 | 0.4 |
| | 4-2 | No trip (stored) | 290 | 6.70 | 21.3 | 58.4 | 87 | 31.8 | 36.5 | 0.7 |
| AB+ | 5 | Before trial | 370 | 8.16 | 22.0 | 62.5 | 77 | 27.0 | 35.2 | 0.1 |
| | 5-1 | Round-trip to Sasebo City (ATR) | 410 | 6.66 | 17.7 | 50.3 | 76 | 26.6 | 35.2 | 0.1 |
| | 5-2 | Round-trip to Kamigoto Hospital (ATR+Type III) | 470 | 6.63 | 17.6 | 50.2 | 76 | 26.5 | 35.1 | 0.7 |
| | 5-3 | No trip (stored) | 570 | 7.85 | 21.0 | 59.5 | 76 | 26.8 | 35.3 | 0.1 |

Abbreviations: ATR, active transport refrigerator; Constar II, RBC Constar II; Hb, hemoglobin; Hct, Hematocrit; MCH, mean corpuscular hemoglobin; MCHC, mean corpuscular hemoglobin concentration; MCV, mean corpuscular volume; Plt, Platelet; RBC, red blood cell(s); Type III, RBC cooler type III; WBC, white blood cell(s).

**Table 3.** Results of the biochemical tests.

| Sample Number and RBC Solution Storage/Transport | | Blood Glucose (mg/dL) | AST (U/L) After Trip (After Fitration) | LDH (U/L) After Trip (After Fitration) | Na (mEq/L) | K (mEq/L) | Lactate (mg/dL) | Ammonia (μg/dL) |
|---|---|---|---|---|---|---|---|---|
| 1 | Before trial | 338.0 ± 1.2 | 8.5 ± 0.6 | 57.5 ± 3.1 | 85.3 ± 0.5 | 49.0 ± 1.0 | 175.8 ± 3.7 | 184.0 ± 15.2 |
| 1-1 | Round-trip to Kamigoto Hospital (ATR) | 291.0 ± 0.8 | 9.5 ± 0.6 (9.3 ± 0.5) | 64.0 ± 2.9 (68.3 ± 2.6) | 83.0 ± 0.0 | 53.6 ± 0.7 | 183.4 ± 5.3 | 239.0 ± 24.4 |
| 1-2 | No trip (stored) | 276.0 ± 0.8 | 10.0 ± 0.0 (11.3 ± 0.5) | 74.0 ± 2.9 (93.3 ± 5.4) | 83.8 ± 0.5 | 53.3 ± 0.6 | 204.6 ± 2.1 | 321.5 ± 17.8 |
| 2 | Before trial | 328.0 ± 2.9 | 7.3 ± 2.5 | 64.3 ± 9.5 * | 87.3 ± 0.5 | 49.3 ± 1.3 | 176.2 ± 6.7 | 234.5 ± 25.3 |
| 2-1 | Round-trip to Kamigoto Hospital (ATR) | 293.5 ± 2.4 | 4.0 ± 0.0 (4.5 ± 0.6) | 40.8 ± 0.5 (42.0 ± 3.2) | 85.3 ± 0.5 | 52.3 ± 0.8 | 183.4 ± 13.6 | 240.3 ± 31.0 |
| 2-2 | No trip (stored) | 263.3 ± 1.0 | 5.8 ± 0.5 (5.0 ± 0.8) | 66.5 ± 5.2 (75.3 ± 5.5) | 83.8 ± 0.5 | 55.1 ± 0.7 | 205.0 ± 4.9 | 347.8 ± 66.2 |
| 3 | Before trial | 412.2 ± 0.5 | 6.3 ± 0.5 | 38.0 ± 0.8 | 86.0 ± 0.0 | 45.4 ± 1.0 | 173.2 ± 3.3 | 228.8 ± 37.0 |
| 3-1 | Round-trip to Kamigoto Hospital (ATR+ Constar II) | 370.0 ± 1.4 | 7.8 ± 0.5 (7.8 ± 0.5) | 55.5 ± 3.7 (52.8 ± 2.5) | 84.0 ± 0.0 | 49.7 ± 0.5 | 156.7 ± 10.3 | 292.5 ± 33.6 |
| 3-2 | No trip (stored) | 361.3 ± 1.3 | 8.3 ± 0.5 (8.8 ± 0.5) | 54.0 ± 2.8 (65.8 ± 5.2) | 84.3 ± 0.5 | 49.0 ± 0.5 | 167.6 ± 5.1 | 276.8 ± 46.5 |
| 4 | Before trial | 342.8 ± 1.0 | 6.5 ± 0.6 | 60.5 ± 3.1 | 86.8 ± 0.5 | 46.8 ± 1.0 | 173.2 ± 3.3 | 232.8 ± 8.4 |
| 4-1 | Round-trip to Sasebo City (ATR) | 303.0 ± 0.8 | 7.0 ± 0.0 (8.0 ± 0.0) | 70.8 ± 2.9 (89.0 ± 5.5) | 83.8 ± 0.5 | 51.6 ± 0.5 | 193.2 ± 8.9 | 263.5 ± 41.4 |
| 4-2 | No trip (stored) | 290.0 ± 0.8 | 7.3 ± 0.5 (7.8 ± 0.5) | 70.0 ± 1.2 (84.8 ± 7.5) | 84.8 ± 0.5 | 51.0 ± 0.7 | 202.8 ± 11.7 | 233.3 ± 59.0 |

**Table 3.** *Cont.*

| Sample Number and RBC Solution Storage/Transport | | Blood Glucose (mg/dL) | AST (U/L) After Trip | LDH (U/L) After Trip | Na (mEq/L) | K (mEq/L) | Lactate (mg/dL) | Ammonia (μg/dL) |
|---|---|---|---|---|---|---|---|---|
| | | | (After Fitration) | (After Fitration) | | | | |
| 5 | Before trial | 394.0 ± 0.8 | 10.0 ± 0.0 | 68.5 ± 1.7 | 78.3 ± 0.5 | 55.8 ± 1.3 | 179.2 ± 5.8 | 175.0 ± 6.5 |
| 5-1 | Round-trip to Sasebo City (ATR) | 356.3 ± 0.5 | 9.3 ± 0.5 (10.0 ± 0.0) | 57.5 ± 1.0 (71.0 ± 3.9) | 77.8 ± 0.5 | 57.6 ± 1.1 | 189.6 ± 2.7 | 260.3 ± 34.4 |
| 5-2 | Round-trip to Kamigoto Hospital (ATR+Type III) | 354.8 ± 1.0 | 14.0 ± 0.0 (14.8 ± 0.5) | 119.8 ± 1.7 (131.8 ± 4.1) | 78.0 ± 0.0 | 57.6 ± 0.8 | 188.8 ± 9.3 | 290.5 ± 44.9 |
| 5-3 | No trip (stored) | 332.8 ± 1.3 | 12.6 ± 0.6 (14.0 ± 0.0) | 94.0 ± 1.8 (125.8 ± 5.7) | 77.8 ± 0.5 | 58.1 ± 0.7 | 198.4 ± 4.1 | 320.5 ± 30.8 |

* The fourth sample was excluded from the analysis. Abbreviations: AST, aspartate aminotransferase; ATR, active transport refrigerator; Constar II, RBC Constar II; K, Potassium; LDH, lactate dehydrogenase; Na, Sodium; RBC, red blood cell(s); Type III, RBC cooler type III.

### 3.5. Evaluation of Hemolysis Using LDH

For the evaluation of hemolysis, we assessed the RBC solution quality, especially for LDH, before and after the journey and after filtration. The state after filtration indicated the state of actual transfusion.

#### 3.5.1. Round-Trip to Kamigoto Hospital (with a UH): RBC Solution in the ATR Round Trip vs. No Trip (Stored at the Laboratory)

Blood packs #1-1 and #2-1 were transported in the ATR to Kamigoto Hospital and back. The mean LDH of the transported RBC solution (#1-1 [64.0 ± 2.9 U/L], #2-1 [40.8 ± 0.5 U/L]) was significantly lower than the mean LDH of the RBC solution in storage (#1-2 [74.0 ± 2.9 U/L]; #2-2 [66.5 ± 5.2 U/L]) ($p = 0.003$ and $p = 0.002$, respectively). The mean LDH values remained the same after filtration ($p = 0.0011$ and $p = 0.000$, respectively). LDH values are presented in Table 3.

#### 3.5.2. Round-Trip to Kamigoto Hospital (with UH Flight): RBC Solution Using RBC Constar IIs for the Round-Trip Helicopter Flight vs. No Trip (Stored at the Laboratory)

In the case of the UH, the method using Constar IIs did not affect the LDH levels (55.5 ± 3.7 vs. 54.0 ± 2.8 U/L; $p = 0.543$), and after filtration, RBC solutions stored at the laboratory had higher LDH values (52.8 ± 2.5 vs. 65.8 ± 5.2 U/L; $p = 0.011$).

#### 3.5.3. Round-Trip to Kamigoto Hospital (with UH Flight): RBC Solution Using RBC Cooler Type IIIs for the Outward Flight vs. No Trip (Stored at the Laboratory)

After the trip, the LDH level of the RBC solution transported with the type IIIs (#5-2) was significantly higher than that of the stored solution (#5-3) (119.8 ± 1.7 U/L vs. 94.0 ± 1.8 U/L; $p < 0.000$). After filtration, the LDH level was not significantly different between the transported and stored samples (131.8 ± 4.1 vs. 125.8 ± 5.7 U/L; $p = 0.14$)

#### 3.5.4. Round-Trip to Sasebo (Left in Sasebo Instead of Going to Kamigoto Hospital; without UH Flight) vs. No Trip (Stored at the Laboratory)

In #4, the transport by air and car to Sasebo had no significant impact on LDH levels and were similar (70.8 ± 2.9 U/L vs. 70.0 ± 1.2 U/L; $p = 0.653$) between RBC solution samples brought back from Sasebo without a helicopter and those stored at the laboratory. After filtration, there was also no difference between the samples (89.0 ± 5.5 U/L vs. 84.8 ± 7.5 U/L; $p = 0.402$).

Lastly, the LDH of, in #5 brought from Sasebo was significantly lower (57.5 ± 1.0 vs. 94.0 ± 1.8 U/L; $p = 0.001$) than that of the sample stored at the laboratory. The findings were similar for the samples after filtration (71.0 ± 3.9 vs. 125.8 ± 5.7 U/L; $p < 0.000$).

The results varied, but overall, the LDH levels of the RBC solutions stored at the laboratory were higher or the same as those of the transported samples. Initially, the RBC

solution sandwiched by type III was found to have LDH levels and showed hemolytic activity, but after filtration, no difference was observed.

### 3.6. Cross-Matching Tests

Cross-matching tests using whole blood samples from the transported segment tubes were not problematic.

## 4. Discussion

We achieved quality control of the RBC solution for transfusion transported to a remote island using a drone. The cross-matching tests were unremarkable. One of the reasons for the success of this trial was that the weather had little effect on the experiment.

To date, there have been no reports on the transport of RBC solution for transfusion at >55 km by a remotely operated UH in Japan. Additionally, transportation of the RBC solution for transfusion by remote control in Japan has not been evaluated.

We would like to discuss our use of UH in this experiment. First, we did not want to use the fixed-wing method of throwing blood out into the air while in flight at the destination. There are also no reports on the visual transport of heavy medicines using the multi-copter. For example, Garcia IQ et al. reported excellent results with the Matrix 300 RTK, which is a multi-copter [19]. However, the Matrix 300 RTK cannot transport heavy-weight objects. In addition, a previous study showed the time-saving advantage of using multi-copter for blood transport [20]. However, we did not perform multi-copter transportation in this study due to weight overload and cold storage. Multi-copters are suitable for short distances around town but not for urgent demands over long distances with temperature control. We always consider temperature control to maintain the quality of the blood. Our study aimed to control temperature, even in multi-copters. However, due to distance, it is difficult to use a multi-copter for maritime transportation and control temperature at the same time.

For transportation, it was also important to clarify the weighing method with ATR, the JRC method, and the temperature control with a simpler method. Therefore, we used UH with mixed shipments as our transportation method.

In summary, the choice of transportation by UH is based on the stable industrial use in agriculture, the ability to carry the ATR700, which weighs >7 kg and can be temperature-controlled, the ability to use conventional temperature control boxes, the stable cruising range, and the us remote control via satellite.

The quality of the RBC solutions after each journey was evaluated in the current study. We previously evaluated regular automobiles [13], bullet trains (data not shown), scheduled passenger aircraft (especially the flight between Haneda and Nagasaki (data not shown) transport temperature, especially when coupled with vibration. We believe that temperature control is the best way to maintain RBC solution quality during transportation, even though this has not been specifically indicated in our data.

In this study, the slight temperature change caused by the opening of the refrigerator door in the laboratory may result in inferior results compared to transport using an ATR in which the temperature was stable. This was also the case with RBC solution transport to Ogasawara [14,15]. A slightly agitated environment might result in higher quality RBC solution than a static environment; however, this was not evaluated in this experiment.

In addition, for effective RBC solution transport in this study, we considered the relationship between the refrigerant, the box, and a reliable ATR. For the temperature management of the ATR, its temperature stability for >24 h during sea transportation to Ogasawara was considered [14,15].

We expected that Constar IIs would result in a stable temperature as they are used by the JRC, and the results met our expectations. However, type IIIs, which we had used in our previous transport experiments, reached unexpectedly high temperatures, resulting in possible hemolysis.

This temperature spike was caused by a change in the type III box from the usual cooler box to box B. Box B was not properly pre-cooled, and during installation under the UH, it was placed close to the hot exhaust muffler, causing a temperature of 6.3 °C upon arrival at the Arikawa port. After the flight, the RBC solution (#5-2) was transferred into the ATR, and the temperature continued to rise up to 7.0 °C because of the residual heat and recovered below 6 °C in 10 min until 11:29 in ATR (data not shown). The temperature inside the transported box B was high at >8 °C before the flight, and up to >9 °C until landing, and it seems that the temperature in the external environment had risen. This might be due to the exhaust tube of the UH or some gap between the Type III and RBC solution, and the temperature rise in the unmeasured area was >6 °C for a long time.

Although the type III results were slightly poor, the results after filtration suggest that the temperature spike may not have substantially affected the RBC solution. In Japan, RBC solutions must be stored at 2–6 °C. The recent change from a 30 min to a 1 h deviation suggests that the increase in temperature in a short period is not a significant problem [21].

Moreover, we detected sustained vibrations in UH, although sudden vibrations are probably greater in vehicles. The vibration did not seem to affect the blood in this study compared to when transported by car or aircraft. However, whether any vibration can cause hemolysis remains unelucidated.

Recently, Oakey A. et al. reported on vibrations in drones for insulin, which is easily degraded, in particular, the difference between the vibration inside the container and the vibration outside the container, which was significantly attenuated by the container [22]. In our study, we only measured the vibration logger attached to the outside of the container which only measures low frequencies; therefore, the extent of the vibration inside the container, especially high-frequency vibrations, which might have a strong impact on the blood, remains unknown. However, even if blood is more affected by the vibration of UH than of other vehicles, the effect is likely to be less than the degradation caused by temperature changes. In a study by Johannessen KA et al., the effects of 10–30 g vibrations for 1–2 h on blood are considered to be a problem [23]. In our study, the sudden vibration of approximately 4 g, and the continuous vibration of about 2 g, showed no significant effect although there appeared to be problem with the logger's vibration frequency sensor.

Since the frequency that the logger can detect is 0–60 Hz and the engine speed at cruise is 6000 rpm (100 Hz), we believe that the actual acceleration is higher than the measured one.

This study focused on the safety of transported blood for transfusion. Blood products are similar to drugs as they are not dangerous substances, while specimens can be dangerous substances [18]. Blood specimens that have not been tested for the hepatitis virus or HIV are potentially more dangerous than drugs and should not be treated the same as blood transfusion products; however, blood products are considered drugs approved by the Japanese Pharmacopoeia and are not hazardous materials that can be transported on regular passenger planes [24].

Even if they are hazardous materials, there is no problem in transporting them by ATR, which is allowed to transport them in the same way that they are normally transported on airplanes because the blood is first protected by a plastic pack, then in a plastic bag, plastic tray, and finally in an outer box that has gaps due to the structure of ATR 700.

In box A, the blood pack is contained in a plastic case with a space and an outer box containing interference material, and blood for transfusion is not a dangerous substance; therefore, there should be no problem. Box A is also used for blood transport by passenger aircraft in the Amami region. Box B, which was created to transport specimens by drone, has a triple-layered structure consisting of a blood pack covered by a plastic bag, an interference space, an air-filled interference material, and an outer shield.

Notably, LDH and AST, especially LDH, were used to assess hemolysis. A transfusion set-pass experiment, unique to us, was used to assess RBC solution quality. We previously reported that RBC solution had hemolytic activity after filtration, which was mainly due to poor temperature control during vehicle transport that may increase the LDH levels in RBC

solutions after transport, even if LDH is not a problem due to transfusion set-pass [25]. This emphasizes the importance of measuring LDH, which passes through the transfusion filter before transfusion in clinical settings. To add about filtering, when blood is transfused, damaged red blood cells are destroyed and LDH rises. It is important to measure LDH after passing through the transfusion set (after filtering) because this means that even a small amount of red blood cell damage can be detected by LDH.

Most of the previous reports on blood transport are based on blood specimens, and there are few studies that evaluate transported blood for transfusion under actual temperature control.

This study had some limitations. As an indicator of hemolysis, companies in Japan do not provide tests for free hemoglobin. Therefore, the JRC itself measures free hemoglobin using an absorbance meter. However, since we did not have access to an absorbance meter, we measured the LDH levels instead, which we believe to be a low-cost and responsive test for hemolysis. For the biochemical tests, the mean was obtained using four samples. Although four samples were sufficient for this study, this may not be sufficient to accurately measure rapidly degrading substances, such as ammonia. However, an increase in the number of samples might reduce the volume of the RBC solution for the transportation trial.

Vibration could not be staged due to the lack of detailed time records for each phase of the flight, including take-off, landing, and cruise flight. This was because the flight time and vibration measurement time were not linked.

Both outbound and inbound flights were not under high winds, but the flight times were slightly different. The manufacturer recommends a wind speed of 10 m/s or less as a possible flight condition for the UH and does not allow it to fly in heavy rain due to wear on the carbon rotors. Strong wind and heavy rain can limit UH flights. As a matter of calculation, the maximum airspeed of the UH was 72 m/h (20 m/s), so if the tailwind is 10 m/s, the ground speed is calculated to be 108 km/h (30 m/s). Moreover, if the headwind is 10 m/s, the ground speed drops to 36 km/h (10 m/s), so excluding the time for takeoff and landing, the flight time with a headwind is three times the flight time with a tailwind.

Additionally, in the event of a disaster, it will be important to transport RBC solutions by UHs to a nearby base. However, UHs may be too large for subsequent transportation to certain locations. This is similar to the disruption of transport that occurred during the COVID-19 pandemic.

Therefore, we think a good way that the UH is used to reach the base where it can be installed, and the small multi-copters are used for short-range transportation to specific locations.

Demonstrations of flights using large uncrewed aerial vehicles to farther, more remote areas are currently being planned, and we believe that this will provide more reliable results.

## 5. Conclusions

We succeeded in the transportation and quality control of the RBC solution for transfusion to a remote island using UHs. We believe that temperature is the most important factor affecting the quality of RBC solutions during transportation.

The UH transport was able to transport blood under controlled temperatures, and the actual analysis of the blood showed that it was possible to meet the rapid demand for blood transfusions anywhere in Japan, including remote areas, which proved our hypothesis. Blood transported at an appropriate temperature can also be used before the expiration date if the blood was not used upon arrival.

This trial is the first to transport packs of RBC solution for transfusion beyond a part of the western Pacific Ocean. Our success indicates novel transportation avenues for remote islands. In the long run, we want to ensure safe and secure transport to supply RBC solutions to remote Japanese islands.

**Author Contributions:** Conceptualization, K.Y.; safety, M.M. and N.H.; demonstration, K.Y. and T.Y.; testing and certification of data, H.F. and M.N.; writing and administration, F.Y. All authors have read and agreed to the published version of the manuscript.

**Funding:** This study was supported by Japan Airlines Corporation.

**Institutional Review Board Statement:** The ethics committee of The Tokyo Metropolitan Bokutoh Hospital approved this project at No. 30-082-3, 7 June 2019.

**Informed Consent Statement:** Not applicable.

**Data Availability Statement:** Not applicable.

**Acknowledgments:** A part of this study was written based on the results of Smart Island Projects promoted by the Ministry of Land, Infrastructure, Transport and Tourism in Japan.

**Conflicts of Interest:** The authors declare no conflict of interest.

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
