# Peer review of "Quality Control of Red Blood Cell Solutions for Transfusion Transported via Drone Flight to a Remote Island"

_drones, doi:10.3390/drones5030096_

Round 1

Reviewer 1 Report

The topic is very interesting, and the results would be useful to extend the use of drones in real-life.

Author Response

Thank you for your careful peer review.

Reviewer 2 Report

This study developed a blood transportation system via drones can meet the rapid demand for blood transfusions anywhere in Japan.
The methods are properly explained and the results of flight experiments are evaluated by the numerical data in temperature, vibration and hematological evaluation.

Title: I think that the number "a distance of 55,000m" is not so important as a title.

Page5: Please increase the resolution of figure 2 because the charachters are hard to read.

Line 356-357 in Page12: Please input as an equation of External force with equation number.

Line 656-357 in Page13: Please refer the equation number in Line 356-357 in Page12 that I mentioned in the previous point. 

Author Response

Response for Reviewer 2

Thank you for your careful peer review. Corrections have been made.

Title: I think that the number "" is not so important as a title.

(Response) Title: Thank you, good advice. I deleted “over a distance of 55,000m”.

Page5: Please increase the resolution of figure 2 because the charachters are hard to read.

(Response) Figure 2; Thanks for pointing it out . I did.

Line 356-357 in Page12: Please input as an equation of External force with equation number.

(Response) Thank you. I changed it.

Line 361-362:  External force(g)=√(X‐axis  force^2+Y‐axis force^2+(|Z‐axis force|-1)^2)(g)

Line 656-357 in Page13:

(Response) Thank you.

Line 381-382:I changed it as same.

Reviewer 3 Report

COMMENTS TO THE PAPER.

I think this paper adds valuable knowledge to the topic of drone transport of transfusion products, and no doubt it qualifies for publishing. However, in its current configuration there are several flaws and some confusing sections. I strongly advise the authors to revise this paper to make it an important contribution to the topic studied.

GENERAL REMARKS.

I would recommend the authors to change some of the terms used. They use a mix of meters and kilometers. The title uses 55,000 meters, in the paper they use 900 000 m for a civil airplane trip which is rather unusual, whereas they use 55 km in distance between cities in the text and in the figures. I recommend using kilometers. Also, they use velocities in m/s, most literature use km/hour. Please clean up this.

I also raise a question regarding the statement that this is the world's first flight of an RBC solution over 55 kilometers distance over the sea. Previous studies have reported similar transports that have been quite longer than 55 kilometers, other studies have flown over sea, although shorter distances. The authors do not explain the significance of 55 kilometers over the sea, and why the paper should be presented with such a label. As I observe it, one of their arguments is that this was done flying out of sight. But flying out of sight for rather long distances is not new, and is not expected to influence biologic quality. They should also explain why it is so “remarkable” to fly over ocean, which it of course may be due to different maritime wind conditions than for example inland conditions. But they should state more related to this.

I am not sure you should use “the world’s first transport in its kind”. I cannot disclose my source, but military transports have performed this earlier, although not as scientific reports. Flying drones at differing distances or locations are not huge innovations any longer, it has been performed in large volumes, and proof of concept in health care is getting better for many use-cases. Therefore, they do not need to “sell” their study as first in the world, I do not consider that important at all. Instead focus on what missing parts in knowledge they have covered.

I also miss a sharper discussion of the limitations of their study. We are presented with the weather conditions during the days of study, being little precipitation and wind of 1 m/s and 2.9 m/s. The authors should discuss the general impact of their findings related to the fact that it was conducted with rather calm and favorable weather conditions during the tests. How would their results relate to a day with 10 or maybe 15 m/s wind and much precipitation? Which often may occur in situations of disasters caused by stormy weather and so on. This is a limitation of their study, not disqualifying in any way, but should be discussed, especially related to their statements in the introduction at line 49, where they tell us that “there have been no conclusive reports about the condition of blood following drone transport”. I do not consider their study conclusive either, considering the limitations in the study. But their study adds valuable aspects which the authors should discuss; how should future research be focused?

The authors do not place their study appropriately in the existing literature, and 8/20= 40 % of their references seem to be from their own works? This is not satisfactory. They have missed much previous literature which actually may be lined up against their own study and strengthen their message. As a few examples that may track them to more literature, these papers may be relevant because they contain multiple useful references:

Zailani, M.A.H., et al., Drone for medical products transportation in maternal healthcare: A systematic review and framework for future research. Medicine, 2020. 99(36).

Moshref-Javadi, M. and M. Winkenbach, Applications and Research avenues for drone-based models in logistics: A classification and review. Expert Systems with Applications, 2021. 177: p. 114854.

Carrillo-Larco, R., et al., The use of unmanned aerial vehicles for health purposes: a systematic review of experimental studies. Global health, epidemiology and genomics, 2018. 3.

MATERIAL AND METHODS

This is a rather complicated section of the paper, and I agree that it must be. However, multiple aspects are repeated, many times. Should be simplified.

This is also valid for the Results section where detailed text of flights is presented despite already being described earlier in the document. Please simplify, make an easier way to connect differing test situations.

Statistics.

It is stated that all analyses were conducted with Excel 2009. But the authors present no statistical methods for their analyzes ending up in p-values. As I understand their presentation, the p-values are assessed within each blood solution, analyzed across 4 test samples from flights/storage of the same solution? What statistical method was used to assess the p-values in this context? Furthermore, it is stated that mean LDH from flights was significantly lower than those in the solution in storage. What does this signify? Then you state that the mean LDH values remained the same after filtration and use P values for tests being similar. What was your methods for these statements and the p-values?

From the data presented, it is obvious that LDH has not increased after flights. I assume an increase would be the problem if it had occurred, but how do we interpret the statement that these values were significantly LOWER than mean values of solutions in storage? Please explain. Actually, your results may simply be interpreted as “there were no significant effects of flying RBC solutions at all”?

These sections should be better explained and discussed more clearly.

RESULTS-

There is a substantial overlap in tables and text. Please do not repeat all details in tables as full text. Focus your text on the main findings in the tables.

Table 3 more or less covers all data in Table 4. Should be simplified to one table, preferably based on Table 3. The reader may easily comprehend the data in Table 4 from Table 3 by some simple explanations and instructions.

The authors are using some inappropriate formulations. An example is at line 358 where they state that “total external forces on the UH were equal to those in an automobile.”

They were not equal, but similar/comparable/same level.

Please observe that if the given flight times and distances in your description is used to calculate velocities, such calculations give slightly different results than you have stated.

Line 302: “the flight time was different on the outward and return trips due to required government permissions and because the speed during the return journey was less than 20 meters per second.”

How could governmental requirements influence flight time? Please explain. As I understand it, all velocities were below 20 meters per second (equivalent to < 72 km/h)? These velocities are not at all decisive, but your presentation may cause some confusion for the reader. I would strongly suggest simplifying the presentation of these data.

DISCUSSION

The authors should start the discussion with a focused description of their main results and message they want to provide. As I observe it, the study showed that flying blood solutions 55 kilometers with an UH did not influence blood solutions negatively. They also compared ground and air transport, which is of interest. Then they should present what new knowledge they will claim to have establish.

In the main part of the discussion, they present a mix of arguments related to multi-copters and helicopters, machine sizes and travel distance. I find this part of the Discussion confusing, and it should be restructured. This part must be sorted out and need to be clarified with respect to the contribution of the current paper. It is a mix of terms of multi-copters and larger machines like their UH. What is the message in this context? On what basis do they lean their statement that “it is difficult to use a multi-copter for more routine transportation and control temperature at the same time period”? This needs a reference.

They also tend to refer to previous research and use terms that can be interpreted as “comparing” results across different studies. Different results across different studies must be done with care. Of note, the difference between dropping blood solutions versus landing on ground is important but must be discussed properly across differing studies.

The second paragraph tells us about parts of the methods that already have been presented previously in the paper. This should be removed.

In the third paragraph at line 499 they state that there have been no reports on drone transport of blood solutions for transfusion beyond 55 kilometers by a remotely operated vehicle. This is not a good formulation, it may be true, but they should rephrase it to discuss long distance flights. What is a long-distance transport and what about 90 kilometers or 150 kilometers? Do results from the current study apply to such distances? A very important perspective.

Line 503: It may possibly be a slip of the tongue when it is stated “I would like to discuss our use”, more adequate phrasing is “We would like to discuss our use”.

Lin 520: they state that there have been no reports of blood solution transport across the ocean using a drone. This is still valid after their study, as they tell us they did not use a drone but a helicopter. Are these equivalent? Discuss.

Line 543: In the paragraphs from line 543 and following you discuss some findings with the type 3 box and relate this to the exhaust muffler placement? Was this placement intended to study the effect of being close to the exhaust muffler/heat, or was it an unexpected consequence of an arbitrary placement at that site?

Line 559: “although the type 3 results were slightly poor, the results after filtration suggests that the temperature spike may not have substantially affected the RBC solution”. Whas this a research target, to see whether such a temperature increase would cause significant changes, or was it an arbitrary happening.

For readers not being experts in biochemistry, it would also be very useful to discuss the significance of analyses comparing values before and after filtration.

In section lines 572-584 you relate your findings to a few previous papers. Here you use g-force in relation to vibration. Vibration is usually measured in Hertz. Please explain why you use g-forces.

Line 576: “However, even if the impact of vibration is stronger in blood than in other vehicles, the effect is likely to be smaller than the degradation caused by temperature changes.”

I do not understand this statement. Stronger in blood than in other vehicles? Gives no meaning.

Line 585. “The degradation of the blood in box B might be caused by vibration”. Needs explanation. speculation?

Line 590: “as mentioned in the materials and methods, there are some controversies in previous studies.”

What is meant by this? Controversies within previous papers, or across previous papers? Anyhow, descriptions of controversies in previous studies or across previous studies should be presented in the introduction/background or Discussion, not in a methods section.

Line 602: “When the bag ruptures”. I assume they mean if the bag should happen to rupture?

Line 609: “notably, LDH and AST, especially LHD, were used to assess hemolysis”. As I observe it, the presentation is mainly based on LDH and that should be focused?

Line 631: “Additionally, in the event of a disaster, it will be important to transport RBC solutions by UHs to a nearby base. However, UHs may be too large for subsequent transportation to certain locations. This is similar to the disruption of transport that occurred during the COVID-19 pandemic. We believe that the use of a small uncrewed multi-copter addresses the challenges of transporting RBC solutions to medical facilities to meet RBC solution demand.

You have previously stated that temperature may not be controlled in such vehicles? Please elaborate.

Line 636: “This trial demonstrated that an automobile, a passenger aircraft, and an UH could safely transport packs of RBC solution. We believe that visual flights with small drones would successfully address RBC solution needs during a disaster.”

There are no reference to why they believe this, and no data in the current study supports such “beliefs”. Maybe this should be rephrased to what more research is needed on this topic?

Author Response

Thank you for your careful peer review. I have made all the corrections I could. I'm sorry that I haven't answered your question completely.The contents are listed in the attached file.
